# Melting Curve Analysis of Aptachains: Adenosine Detection with Internal Calibration

**DOI:** 10.3390/bios11040112

**Published:** 2021-04-08

**Authors:** Chenze Lu, Christine Saint-Pierre, Didier Gasparutto, Yoann Roupioz, Corinne Ravelet, Eric Peyrin, Arnaud Buhot

**Affiliations:** 1College of Life Sciences, China Jiliang University, Hangzhou 310018, China; chenzelu@cjlu.edu.cn; 2University Grenoble Alpes, CEA, CNRS, IRIG, SyMMES, F-38000 Grenoble, France; christine.saint-pierre@cea.fr (C.S.-P.); didier.gasparutto@cea.fr (D.G.); yoann.roupioz@cea.fr (Y.R.); 3University Grenoble Alpes, CNRS, DPM, F-38000 Grenoble, France; corinne.ravelet@univ-grenoble-alpes.fr (C.R.); eric.peyrin@univ-grenoble-alpes.fr (E.P.)

**Keywords:** split-aptamers, small molecule detection, aptachain self-assembly, melting temperature, calibration/normalization

## Abstract

Small molecules are ubiquitous in nature and their detection is relevant in various domains. However, due to their size, sensitive and selective probes are difficult to select and the detection methods are generally indirect. In this study, we introduced the use of melting curve analysis of aptachains based on split-aptamers for the detection of adenosine. Aptamers, short oligonucleotides, are known to be particularly efficient probes compared to antibodies thanks to their advantageous probe/target size ratio. Aptachains are formed from dimers with dangling ends followed by the split-aptamer binding triggered by the presence of the target. The high melting temperature of the dimers served as a calibration for the detection/quantification of the target based on the height and/or temperature shift of the aptachain melting peak.

## 1. Introduction

The detection of small molecules plays an important role in various fields like food safety, environmental control, diagnosis, etc. [1,2,3,4,5]. Antibodies, the most commonly used probes in biosensors, are generally difficult to raise against such small targets, i.e., with molecular weight lower than 1000 Da. On the other hand, aptamers, single stranded oligonucleotides specifically selected against their target by the Systematic Evolution of Ligands by EXponential enrichment (SELEX) method, are interesting alternative probes [6,7]. Due to their in vitro selection method, ease of synthesis and chemical functionalization, low cost and simple incorporation in biosensors, aptamers have emerged as a new and competitive recognition element for various applications in the past two decades [8,9,10,11,12,13]. In the particular case of small molecule detection, the advantages of aptamers over antibodies are even more crucial [14,15,16,17,18,19]. Due to their reversible folding in a particular 3D conformation, they exhibit a binding pocket with strong affinity and selectivity towards small targets while the large size of antibodies compared to the targets confers them a comparative disadvantage.

Nonetheless, the small size of the target molecules and their low concentrations in the solution represent the main challenges in developing biosensors for their detection. Indirect signal amplification is often required in the development of novel label-free sensing methods due to the small signal obtained from the direct binding of the small targets to the probes. Nanoparticle-based strategies are often used to circumvent this problem. Many examples can be found in the literature where molecular probes are covalently bound or adsorbed to metal (usually gold) substrates or nanoparticles [20,21,22,23,24,25,26,27,28,29]. However, the preparation of such molecular architectures involves complex functionalization procedure and has a high cost in the production and modification of the nanoparticles. Recently, questions arose about the use of adsorbed adenosine aptamers on nanoparticles or graphene as a biosensor strategy [30]. Furthermore, in homogenous phase detection, any change in the probe concentration and/or in sample content (pH, salt concentration...) generally led to variability in the detection method [31,32,33]. We propose a homogenous phase detection method in which aptamer probes are not immobilized on any surface but still provide a noticeable signal for detection with an internal control suitable for calibration and/or normalization.

The detecting system is based on the self-assembly of 1D DNA chains formed by bi-functional oligonucleotides forming double stranded dimers on one hand and split-aptamer sequences on the other hand. The use of the hybridization chain reaction (HCR) forming such self-assembled 1D DNA chains for signal amplification has been widely developed in recent years [34,35,36,37,38,39,40]. However, its use with aptamer recognition to form what are called aptachains, is still in its infancy [3,41,42,43,44]. In our case, the adenosine target served as the trigger for thermodynamic control of the linear DNA nanostructure formation in solution.

In order to follow the various steps of the aptachain self-assembly, we focused on melting curve analysis obtained from UV absorbance at 260 nm. Thanks to nucleic acid’s hyperchromic property, UV monitoring has been extensively used to determine the melting profile of DNA duplexes or to prove the existence of targets [45,46,47,48]. The change in the wavelength of the absorbed light is also used to analyze the interactions that take place in the solution [49,50,51,52,53]. In our method, the melting curve of the aptachain structures obtained from UV spectroscopy is used to detect the change in the binding strength of the split-aptamers with the target. The melting peak and temperature of the split-aptamer bindings are shown to be directly related to the amount of adenosine target present in the solution, thus, allowing its detection and potential quantification. UV spectroscopy is commonly used as a supplementary method to confirm the change in the binding strength, but it is rarely proposed as a detection method because measurements are influenced by the buffer composition or aptamer concentration. For this reason, the reading of the melting peak or temperature could not determine the presence of the target directly. In our study, the building of the aptachains from oligonucleotides containing hybridizing dimer moieties provided an internal reference with a well-defined high melting temperature. Thus, at low temperature, the targets trigger the split-aptamer self-assembly and, at high temperature, the hybridization of the dimers serves as an internal reference providing two different peaks in the melting curve analysis. This approach has enabled the direct detection and quantification of the target.

## 2. Materials and Methods

### 2.1. Reagents and Oligonucleotides

The reagents used for preparing the buffer as well as adenosine and guanosine were purchased from Sigma-Aldrich (Saint Quentin Fallavier, France). The oligonucleotide sequences (see Table 1) were purchased from Eurogentec (Angers, France). The buffer in which the oligonucleotides were mixed consisted of 10 mM HEPES, 5 mM MgCl_2_, and 150 mM NaCl, and its pH was set at 7.4 with HCl and NaOH.

### 2.2. Oligonucleotides Sequence Design

We tested five sequence designs with different binding strengths for the split aptamers as well as a full aptamer as reference. The number of base pairs hybridized in the split-aptamer dangling ends controlled their binding strengths. The DNA sequences near the binding pocket of the split-aptamer are shown in Scheme 1. The sequences are named by the number of complementary base pairs in the split-aptamer dangling ends (blue sequences in Table 1) next to the binding pocket (black sequences). For example, SA3-5A/B illustrated in Scheme 1 had three base pairs hybridized on the left side and five on the right side (blue sequences in Table 1) of the binding pocket. While SA stands for split-aptamer, the final letters A or B relate to an additional Zip sequence (red sequences in Table 1) on the 3′ end, which are complementary to each other and allow the dimer formation (see illustration in Scheme 2). The detailed sequences of these five designs are displayed in Table 1 along with the Apta6 sequence corresponding to a full adenosine aptamer, which will be used as a reference. The binding strength near the adenosine pocket of the five sequence designs is expected to increase with the number of hybridizing base pairs. The binding of the split-aptamer will be called an ‘aptamer bridge’ in the following. To decrease the affinity of the aptamer bridge of the least stable sequences, a non-complementary GAG sequence has been added (Yellow in Scheme 1).

### 2.3. UV-Vis Measurements

The UV measurements were performed on a Cary 100 UV-Vis Spectrophotometer (Agilent, Santa Clara, CA, USA). The sample solutions were obtained by mixing oligonucleotides and adenosine in the buffer solution. The concentrations of oligonucleotides were kept at 0.9 μM to maintain a high resolution in the absorption curve (except when otherwise quoted). For each sample solution containing oligonucleotides and adenosine, a reference sample solution was considered by mixing the same amount of adenosine in the buffer without the oligonucleotides. For UV measurements, 1 mL of sample and reference solutions were injected inside two different cuvettes. The cuvettes were sealed and placed into the chamber slots. UV light at 260 nm passed through the window of the cuvettes and was analyzed to provide the absorption data. The UV light adsorption was determined by subtracting the absorption of the reference sample from the solution sample to eliminate the influence due to the adenosine absorption at 260 nm. Adenosine concentrations were varied based on the purpose of the tests. Guanosine molecules were used as a negative control to analyze the selectivity [27,54].

Before each melting curve analysis, the samples were heated to 95 °C for 5 min and cooled down at room temperature for at least 30 min. During the analyses, a venting system was running to refresh the air inside the chamber and reduce frosting at low temperature. Samples were kept at 15 °C for 10 min before the temperature scan was performed. Then, the solutions were heated up to 80 °C step by step. For every +0.2 °C increase in the temperature, the absorbance of UV light was recorded after 1 min delay. Once the temperature reached 80 °C, the samples were heated to 90 °C with larger increasing steps of +0.5 °C. The temperature decrease followed similar temperature steps. The whole temperature cycle was repeated 4 times for each sample. The different heating/cooling rate at high temperature reduced the evaporation of samples and eliminated its influence on the accuracy and repeatability of the measurement (see experimental data of the 4 temperature scans in Appendix A). The melting curves were obtained by taking the first derivative of the 260 nm UV absorbance as a function of temperature. The peak height of the highest melting temperature was set to one for normalization.

## 3. Results and Discussion

### 3.1. Aptachain Formation and Melting Curve Analysis through Temperature Scans

Split-aptamers are obtained from the original anti-adenosine hairpin aptamer (see Apta6 sequence in Scheme 1) by removing a thymine in the middle of the loop [27,28,29]. This splitting does not strongly affect the affinity towards adenosine since the binding pocket in the middle of the stem is not affected. The split-aptamers (blue in Scheme 2) were combined with oligonucleotide Zip (red). A 24-mer Zip sequence was added to the 3′ end of the split-aptamer while its complementary sequence was added to the other half of the split aptamer couple in order to induce dimer formation (transition 1 to 2 in Scheme 2) at high temperature.

By decreasing the temperature further, the dangling ends of the dimers hybridize to form bridges with the pocket available for adenosine binding. Without adenosine in the solution, the formation of those bridges leads to the formation of aptachains (transition from region 2 to 3 in Scheme 2). In the presence of adenosine, its binding to the pocket stabilizes the bridges and thus the aptachain formation. Indeed, we expect that by increasing the adenosine concentration, a larger amount of targets will bind to the aptamer pocket leading to an increase of the melting temperature (transition from region 2 to 4′ in Scheme 2). At low adenosine concentrations, we may expect that the adenosine binding will occur at temperatures below the formation of the aptachains (transition from region 3 to 4) thus not affecting the experimental value of the melting temperature for aptachain formation. However, in this case, we still expect that the binding of the targets will affect the melting peak. More importantly, the melting peak for the dimer formation should not be affected by the presence of adenosine and, thus, it may serve as an internal control for normalization of the melting curves.

### 3.2. Impact of Split-Aptamer Design on Aptachain Formation

In order to explore the aptachain formation, we designed five different split-aptamer couples along with similar Zip sequences (SA3-5A/B, SA3-6A/B, SA5-5A/B, SA5-6A/B, and SA5-8A/B). We expected to enhance the stability of the aptamer bridges by increasing the number of hybridized base pairs. The melting curves were obtained for those five designs along with the reference aptamer (Apta6) without adenosine (Figure 1). In order to assess the impact of the presence of adenosine, we also analyzed the melting curves in the presence of a large excess of the target (100 µM). In all cases, oligonucleotides concentrations were set at 0.9 µM. In each of the cases, four different scans were carried out to estimate the reproducibility. No evaporation effect was observed (Appendix A).

First, as expected, we observe a similar high temperature melting peak for all five designs at T_m_ = 75 ± 2 °C, which is compatible with the dimer formation. Indeed, the melting temperature for the Zip sequences is expected at 77.3 °C by the Mfold software [55]. Furthermore, for the full aptamer, the melting peak at high temperature is absent. Secondly, a first melting peak for most of the designs is observed at a lower temperature and is reminiscent of the aptachain formation. Only the dimer SA3-5A/B does not display the first peak corresponding to the aptachain formation in absence of adenosine. The aptamer bridge for this design, which is the least stable of all the designs considered, may explain this lack of aptachain formation. In fact, we added a non-complementary GAG on both sequences in order to further destabilize the aptamer bridge. What is interesting to notice is that the presence of adenosine has an important impact for this particular design since a low temperature melting peak is observed at T_m_ = 28 °C in presence of 100 µM of adenosine. We also performed gel electrophoresis with a large amount of adenosine (1 mM) without noticing any chain formation (see Appendix A). We may expect that the aptamer bridges are not sufficiently stable to support gel migration since the formation of DNA chains was confirmed by gel electrophoresis for self-complementary dimers (Appendix A).

In general, the first melting peaks were affected by the binding strength of the aptamer bridge. The higher the number of hybridization base pairs (hbps), the larger the corresponding melting temperature: T_m_ = 33.3 +/− 1 °C for SA3-6A/B (9 hbps), T_m_ = 40.5 +/− 1 °C for SA5-5A/B (10 hbps), T_m_ = 44.5 +/− 1 °C for SA5-6A/B (11 hbps), and T_m_ = 51.5 +/− 1 °C for SA5-8A/B (13 hbps). As a comparison, the melting curve of the full aptamer only showed one melting peak regardless of the presence of adenosine. The melting temperature T_m_ = 57.5 +/− 1 °C for the aptamer Apta6 is higher than for the binding of split-aptamers (even though the number of hpbs is higher for the dimer SA5-8A/B). This may be explained by the fact that in the case of the full aptamer, the melting corresponds to the folding/unfolding of a single oligonucleotide hairpin. For split-aptamers, the hybridization of two different oligonucleotide sequences implied an additive entropic effect explaining the lower melting temperature. For further studies, it would be interesting to analyze the effect of the loop size in the full aptamer. Most of the studies considered the original sequence with only a single thymine in the loop. However, larger loops may lead to lower hairpin melting temperatures and the increased flexibility of the loop could have an impact on the affinity of the aptamer. When comparing with the full aptamer reference, another difference concerns the shift in melting temperature when introducing adenosine. This shift ΔT_m_ is small (less than 1 °C) for the full aptamer compared to the aptachain structures (ΔT_m_ = 5.5 °C for SA3-6A/B, ΔT_m_ = 2.5 °C for SA5-5A/B, and SA5-6A/B, ΔT_m_ = 2 °C for SA5-8A/B, while not measurable for SA3-5A/B due to the absence of melting peak without adenosine). Moreover, in the case of the aptachain structure, a reliable increase in the intensity of the first peak with the presence of 100 μM adenosine was observed after the normalization calibration by the second peak. In the case of the full aptamer, the direct comparison between the intensities of the single peak with and without adenosine may be hampered by slight changes in solution content and aptamer concentration due to the lack of internal calibration.

Interestingly, the presence of adenosine has an effect on both the maximum height and the melting temperature of the first peak. In all cases, the increase in the maximum height and melting temperature was observed to be more important for the lower aptamer bridge stability. It was evident for the lowest stability (SA3-5A/B) since the first peak was not observed (even for temperature as low as 5 °C) in the absence of adenosine. Furthermore, the relative increase in both maximum height and melting temperature seemed to decrease with the DNA architecture increasing stability. We may explain this observation by the fact that the stability brought by the adenosine binding to the pocket is independent of the dimer designs (since the pocket is not modified by the various designs) while, on the contrary, the hybridization stability increased with the number of hbps present (thus reducing the relative effect of the presence of adenosine).

Based on those observations, the optimal design seemed to be SA3-6A/B since it presented the highest shift in melting temperature and maximum height variation while introducing adenosine in the solution. The design SA3-5A/B could also be of interest since the first peak is clearly absent without adenosine but appears in its presence. However, the first melting peak is observed at low temperatures, which require working with a larger temperature range from 5 °C to 90 °C, rendering the experiment more time-consuming and noisier. The noise observed at low temperature may come from two different effects: the salt precipitation of the solvent inside the cuvette and the condensation outside the cuvette. Thus, the following set of experiments was focused on the SA3-6A/B design.

### 3.3. Impact of Adenosine Concentration on Aptachain Formation

In this section, we focused on the SA3-6A/B dimer design and analyzed the evolution of the first melting peak as a function of adenosine concentration in order to understand the aptachain formation thermodynamics and to determine how it may be used for adenosine quantification. The first melting peak was strongly affected by the increasing concentration of adenosine (for c = 0, 5, 10, 20, 50, and 100 µM see Figure 2) while the second peak could serve as a normalization (no apparent change of the melting temperature). Furthermore, guanosine was used as a negative control to assess the selectivity. The melting curve for 100 µM guanosine lies between the melting curves without and with 5 µM of adenosine confirming the selectivity of the adenosine (split-)aptamer.

As seen in Figure 2, not only the shift in the melting temperature of the first peak, but also its height may be used to determine the presence of adenosine in the solution. The heights of the first peak are presented as a function of adenosine concentration in Figure 3 Left. A continuous increase of the first peak is observed with the increase in adenosine concentration c, leading to saturation at high concentration. We may relate such an increase to the amount of aptamer bridges induced by adenosine. By fitting with a Langmuir model P = P0 + ΔPmax c/(K_D_+c) with the peak height P and the fitting parameters: P0, the height without adenosine, ΔPmax, the maximal shift in height value, and K_D_ the dissociation constant, we determined K_D_ = 19 +/− 2 µM. As expected, this value is slightly higher than the full aptamer dissociation constant (K_D_ = 7 µM) due to the splitting of the aptamer into two strands [14,27]. Still, the difference is minimal illustrating the low impact of the splitting on the binding recognition. The signal to noise ratio for the concentration 5 µM of adenosine is above 3 (with a noise in the peak height estimated around 0.005). However, the signal observed with 100 µM of guanosine (blue curve on Figure 2) is close to the signal obtained for 5 µM of adenosine. Thus, we cannot expect to selectively detect adenosine below 5 µM. On Figure 3 Left, the first peak maximum is close to saturation for adenosine concentration of 100 µM, thus leading to a dynamic range between 5 to 100 µM. The limit of detection (LOD = 5 µM) of our biosensor is comparable with similar homogenous phase detection for aptamer-based small molecule detection without amplification [21,23,54,56,57,58].

Now, let us focus on the temperature shift (Figure 3 Right). When the concentration of adenosine was below 20 µM, the melting peak showed no clear temperature shift while the height of the peak was clearly increasing. Indeed, the melting temperatures observed up to 20 µM of adenosine lie within the error bar of the one observed without adenosine (blue band in Figure 3 Right). However, the melting temperature shifted by +3 °C and nearly +6 °C, respectively for 50 µM and 100 µM of adenosine. In the meantime, the maximum peak height increased linearly from 0.18 to 0.35 when the concentration of adenosine increased from 0 to 20 µM and finally reached 0.44 with the presence of 100 µM adenosine in the solution. Thus, the increase of the peak height was more pronounced at adenosine concentrations lower than 20 µM. In conclusion, the shift in peak height is clearly more relevant to detect and quantify the adenosine concentration at low adenosine concentration than the shift in melting temperature. This indicates once again the importance of a normalization procedure to consistently and precisely measure the peak height.

Interestingly, the melting temperature effect observed as a function of the adenosine concentration qualitatively follows the behavior illustrated in Scheme 2. We may expect that two competing melting/binding temperature-dependent events are involved. First, the aptamer bridges formation occurs at a melting temperature T_m_(bridge) independent of the concentration of adenosine. Secondly, the binding of adenosine within the aptamer pocket is obviously influenced by the adenosine concentration c(Ade), but is also temperature dependent through the dissociation constant K_D_(T). In fact, we expect that half of the bridges are bound by adenosine when, K_D_(T) = c(Ade) defining an adenosine concentration-dependent melting temperature T_m_(Ade) which is logarithmically increasing with the adenosine concentration (illustrated in red in Scheme 2). Indeed, at low adenosine concentration, the melting temperature T_m_(bridge) of the aptamer bridge is higher than T_m_(Ade) at which the adenosine would bind to the aptamer pocket. Thus, the observed melting temperature T_m_(bridge) is constant while the binding of adenosine below this temperature may explain the increase in peak height. With higher concentration of adenosine, the increasing melting temperature T_m_(Ade) is observed due to the stabilization of the aptamer bridges by the binding of adenosine. We effectively observed a linear increase of the melting temperature as a function of the logarithm of the adenosine concentration (Figure 3 Right). The overlap between the two different temperature regimes occurs at c(Ade) = K_D_(Tm(bridge)) = 20 µM, which is consistent with the K_D_ value extracted from the Langmuir fit (Figure 3 Left).

### 3.4. Influence of the Salt Concentration

The benefit of the internal reference in the detection method lies in the reduction of the influence of the buffer composition. In order to prove this point, experiments were carried out comparing the UV-Vis spectroscopy data obtained with different buffers. To begin with, we used a buffer with the same preparation protocol but not in the same batch and not on the same day. In the first set of tests, the two peaks were at 33 °C (T_m1_) and 77 °C (T_m2_) without adenosine. The first peak shifted to 38 °C with 100 µM of adenosine in the solution while the second peak remained the same. The value of all peaks had a 1 °C difference compared to the second set of tests. The peaks without adenosine were at 34 °C (T_m1_) and 78 °C (T_m2_), and the first peak shifted to 39 °C with 100 µM of adenosine in the solution. Slight differences in the buffer resulted in different melting temperatures measured, but the difference between the two melting peaks remained the same. The results supported our claim that the internal reference is an important element enabling the detection of adenosine regardless of the minor difference produced in the preparation of the buffer.

Furthermore, we prepared buffers with different salt concentrations to see how it affected the melting profiles. The 10× buffer was diluted into three different concentrations: 0.5× buffer, 1× buffer, and 1.5× buffer, respectively. Different samples with different concentrations of adenosine (0 and 100 µM) and buffer concentrations were prepared. In order to observe the interest in the normalization, melting curves were represented in Figure 4 with the dimer peak normalized to one as a function of the difference temperature ΔT = T_m_(dimer) − T. The value of the peak corresponding to the dimer formation was thus set at ΔT = 0 °C with 100 µM of adenosine or without adenosine. We observed that independently of the buffer concentration, with the presence of 100 µM of adenosine in the solution, the difference between the two melting peaks was kept constant at 39 °C. The height of the peaks was also similar, although a slight difference was observed for the low salt concentration (0.5×). The exact values of these melting peaks were not the same before normalization, which suggested that the melting temperatures were affected by the salt concentration in the buffer as commonly observed for oligonucleotide hybridization assays [59,60,61]. Thus, the detection was still reliable thanks to the internal reference. We further noticed that the difference between the two melting peaks without adenosine was increased to 46 °C for 0.5× buffer compared to 44 °C for 1× buffer and 1.5× buffer. The larger difference between the two peaks suggested that the binding strength was weaker in the case of 0.5× buffer without adenosine. A possible reason is that the lack of salt led to unsaturated formation of aptamer bridges increasing the effect on its melting temperature (right peaks in Figure 4). On the contrary, the other two buffers contained enough salt to saturate the formation of aptamer bridges. Thus, the internal reference may rule out the influence caused by differences in the buffer concentration, but the salt concentration needs to be high enough to saturate the formation of aptamer bridges.

### 3.5. Influence of Oligonucleotide Concentrations

The concentration of the oligonucleotide strands was another parameter affecting the melting curves. Figure 5 Left presents the melting curves before calibration. The two melting peaks exhibited a very small shift with the change of strand concentrations. On the other hand, the heights of both peaks strongly increased with the oligonucleotide concentrations. We observed increasing noise for samples with lower oligonucleotide concentrations. Their melting curves were smoothed with the Savitzky–Golay method to reduce the noise [62]. In contrast, for higher concentrations, there is a risk of saturating the signal. Thus, as mentioned in the protocol, the oligonucleotide concentration was preferable at 0.9 µM in order to have the best resolution in the melting curve of the aptachain structures. By comparing the melting profile after the normalization of the second melting peak (Figure 5 Right), we noticed that the heights of the first peak were similar to each other. Thus, the normalization proposed by the internal control may correct the height variations of the first melting peak observed due to the variation in concentration of the oligonucleotides.

## 4. Conclusions

In this study, we analyzed the formation of aptachains composed of dimers with split-aptamers towards adenosine as dangling ends. Two different melting temperatures leading to the formation of the aptachains at low temperatures were determined from UV absorbance at 260 nm. The higher melting temperature was related to the dimer formation while the lower melting temperature was representative of the aptamer bridges. Due to the target recognition and further stabilization of the aptamer bridges, the lower melting peak was shown to be target concentration-dependent. Various aptamer sequences were considered to select the most sensitive for the detection of the aptamer target adenosine. Thus, we developed a homogenous phase biosensor for the label-free detection of small targets based on the UV melting curve analysis of aptachains. We obtained a limit of detection of 5 µM which is similar to previous studies without amplification.

While in this study, we considered UV absorbance at 260 nm to reveal the aptachain formation and to characterize the melting curves, other homogeneous techniques could be used based on fluorescence of intercalating dies [63], fluorescence polarization or anisotropy [56,64,65,66,67] to mention a few. Heterogeneous assays may also be considered to allow for simple multiplexing of target detection. For example, melting curve analysis of multiple oligonucleotide probes in parallel was demonstrated with the use of surface plasmon resonance imaging [28,60,68,69,70]. Recently, the formation of aptachains on surfaces was demonstrated by such a transduction technique [41]. It is also interesting to notice that cooperativity effects may sharpen the melting in case of comb-like DNA polymers or nanoparticles [71,72]. It would be interesting to understand if such structures could have an effect on the detection limit of the biosensors.

## Data Availability

Data are accessible upon request to the authors.

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
