# Peer review of "Melting Curve Analysis of Aptachains: Adenosine Detection with Internal Calibration"

_biosensors, 2021, doi:10.3390/bios11040112_

Round 1
Reviewer 1 Report
The manuscript describes the experimentation regarding 5 sequences of Oligonucleotides and a control, as an entrapment method for the detection of adenosine. All experiments were carried out based on thermal analysis and once most suitable aptamer-chain was selected, the evaluation of further parameters as salt concentration, oligonucleotide concentration, adenosine concentration was carried out.
The manuscript proposed a understandable parameter evaluation; however, some comments and questions raised when reading
- The manuscript describes a method for quantification of the interaction between a aptachain and adenosine, that is later read by standard UV-VIs equipment. Therefore, the target of the manuscript shall be more directed towards the molecular interaction, rather than as a biosensor.
- The study only described the interactions and evaluation using an stand alone technique, UV-VIS; therefore, it is recommended to use alternative techniques to verify the described interactions and results.
Reviewer 2 Report
It is very interesting paper and worth to publish in Biosenros. Well written and easy to understand. However, authors should discuss their results with the similar studies and underline the benefits of this method more clearly. Some specific comments:
- The scheme 2 on the left is somehow not clear. Maybe it is possible to rearrange this scheme.
- The description of Figure 4 is not clear. There are 6 melting curves and the description contains 5.
Reviewer 3 Report
In this manuscript, the authors propose a method for adenosine detection with internal calibration. While their study should be of high interest to many potential applications, it lacks some critical issues both in terms of experimental design and representation. I would recommend improvement in experimental designs, analysis and presentation before I can recommend it for publication as an article in Biosensor. Here are some of my key concerns:
- Please illustrate in the figure and the text what is the significance of the golden sequence in SA3-5A. Why do the authors have a different design compared to the other ones?
- The Apta-6 sequence seems sterically constrained with only one T loop at the end! Did the authors try the same aptamer with less steric constrains?
- Scheme-2 is unnecessary and adds very little to the story. First of all, should the y-axis be melting temperature or the absolute value of the temperature? Moreover, using lines to represent different regions (area) is very confusing. How do the authors distinguish between these regions experimentally? For example, can they differentiate between 4 and 4’? The cartoon is also confusing! Shouldn’t there be double strand formation in both side of the aptamer?
- Figure 1: Melting curve is not a physical entity that can be plotted as an axis. The authors should consider showing at least one raw data where they show Temperature vs Absorbance plot and then explain how and why they chose the plot the derivative of the absorbance and name it like wise. The raw data is also important to understand the dynamics of complex formation. Do the authors see any kind of hysteresis for example?
- Figure 1/e: Why the melting temperature of the aptamer much higher than all other cases? Why is there virtually no stabilization when adenosine is added?
- Do the authors see any cooperativity of adenosine binding? If so, they should use Hill-equation for fitting.
- The authors should comment about what is the limit of detection and the dynamic range of their measurements.
Round 2
Reviewer 1 Report
The authors have partially answered the questions, but the information provided sheds light on a better understanding of the phenomena.
Reviewer 3 Report
The authors have incorporated most of my comments. I recommend for publication in biosensor.